# Productivity, efficiency, and overall performance comparisons between attendings working solo versus attendings working with residents staffing models in an emergency department: A Large-Scale Retrospective Observational Study

**Richard D. Robinson**[1,2]**, Sasha Dib**[1]**, Daisha Mclarty**[1]**, Sajid Shaikh**[3]**, Radhika Cheeti**[3]**, Yuan Zhou**[4]**, Yasaman Ghasemi**[4]**, Mdmamunur Rahman**[4]**, Chet D. Schrader**[1]**, Hao Wang**[1] *

**1** Department of Emergency Medicine, Integrative Emergency Services, John Peter Smith Health Network, Fort Worth, TX, United States of America, **2** University of North Texas Health Science Center, Fort Worth, TX, United States of America, **3** Department of Information Technology, John Peter Smith Health Network, Fort Worth, TX, United States of America, **4** Department of Industrial, Manufacturing, & Systems Engineering, The University of Texas at Arlington, Arlington, TX, United States of America

* hwang@ies.healthcare

## Abstract

### Background and objective

Attending physician productivity and efficiency can be affected when working simultaneously with Residents. To gain a better understanding of this effect, we aim to compare productivity, efficiency, and overall performance differences among Attendings working solo versus working with Residents in an Emergency Department (ED).

### Methods

Data were extracted from the electronic medical records of all patients seen by ED Attendings and/or Residents during the period July 1, 2014 through June 30, 2017. Attending productivity was measured based on the number of new patients enrolled per hour per provider. Attending efficiency was measured based on the provider-to-disposition time (PDT). Attending overall performance was measured by Attending Performance Index (API). Furthermore, Attending productivity, efficiency, and overall performance metrics were compared between Attendings working solo and Attendings working with Residents. The comparisons were analyzed after adjusting for confounders via propensity score matching.

### Results

A total of 15 Attendings and 266 Residents managing 111,145 patient encounters over the study period were analyzed. The mean (standard deviation) of Attending productivity and efficiency were 2.9 (1.6) new patients per hour and 2.7 (1.8) hours per patient for Attendings

**Data Availability Statement:** Data cannot be shared publicly because data include patient information. Data are available from the John Peter Smith Health Network, Office of Clinical Research (contact via Dr. Melissa Acosta, email: research@jpshealth.org) for researchers who meet the criteria for access to confidential data.

**Funding:** The authors received no specific funding for this work.

**Competing interests:** The authors have declared that no competing interests exist.

working solo, in comparison to 3.3 (1.9) and 3.0 (2.0) for Attendings working with Residents. When paired with Residents, the API decreased for those Attendings who had a higher API when working solo (average API dropped from 0.21 to 0.19), whereas API increased for those who had a lower API when working solo (average API increased from 0.13 to 0.16).

## Conclusion

In comparison to the Attending working solo staffing model, increased productivity with decreased efficiency occurred among Attendings when working with Residents. The overall performance of Attendings when working with Residents varied inversely against their performance when working solo.

## Introduction

Emergency Department (ED) provider productivity and efficiency are two important performance measures. Productivity is viewed as the number of patient encounters per provider per unit time. It can also be measured by number of Relative Value Units (RVUs) generated per provider per unit time or encounter. Efficiency refers to the time and resources required to complete an ED patient encounter. ED length of stay (LOS), defined as the total time spent within the ED for a given patient encounter, and provider-to-disposition time (PDT), defined as the interval starting with initial provider involvement and ending with disposition selection for a given patient encounter, are among recognized efficiency metrics [1–4]. However, both are affected by multiple factors including ED crowding, patient acuity, and supervision of Residents. [5–7].

Most of the care delivered in training institutions occurs via the Attending oversight of Resident care model [8]. Mixed results are documented regarding provider productivity and efficiency of the Attending-Resident Team as a function of relative team clinical experience [9–11]. A study focused on productivity used the number of new patients per hour seen by either residents or Advanced Practice Providers (APPs) in a Fast Track area (i.e., low acuity patient care area), they found Resident productivity was less than that of the APPs' indicating resident in training might affect their productivities [12]. Previous studies revealed prolonged LOS was experienced by patients receiving care in the Attending oversight of Resident model indicating decreased provider efficiency in the training institution setting [9,10]. However, another study determined that ED LOS was not significantly affected by the presence or total number of trainees in the ED [11]. At present, only a few studies compare provider productivity and efficiency between Attendings working solo versus Attendings working with Residents and none of them examine the differences in productivity and efficiency of individual Attendings within these two groups.

Productivity and efficiency are often consistent when used to evaluate provider performance (i.e., efficient providers are also productive or vice versa). However, it is not uncommon to observe some inconsistencies (e.g., high productivity but low efficiency), thus creating a significant challenge to understanding overall provider performance. To overcome this challenge, prior research introduced a composite index, which was calculated by combining productivity and efficiency [13]. However, a major shortcoming of said calculation is the lack of external validation.

To gain a deeper understanding as to whether working with Residents affects Attending performance within an academic environment, we aim to measure productivity, efficiency,

and overall performance of Attendings and further compare the differences among Attendings working solo versus Attendings working with Residents at an individual level.

## Methods

### Study setting and design

This is a single center retrospective observational study. The institutional review board of John Peter Smith Health Network approved this study (IRB No. 010713.004ex) with the approval of the waiver of the written informed consent. The study hospital is a tertiary referral center located in an urban area serving a community of approximately 2 million. It is a publicly funded hospital. It is also a regional Chest Pain Center, Comprehensive Stroke Center, and a Level 1 Trauma Center. The hospital ED sponsors a 3-year EM Residency Program and managed approximately 120,000 annual visits during the study period. The majority of our patients have no commercial insurance coverage. Approximately 15–20% of patients are covered by Medicare and Medicaid. Approximately 25% of patients are Hispanic.

The ED has dedicated Fast Track and main ED areas. APPs staff Fast Track and see mainly low acuity patients (i.e., Emergency Severity Index [ESI] Level 4 and 5 and low risk Level 3, see detail explanation of ESI in S1.1 Appendix) requiring minimal oversight (< 5%) by Attendings. Residents see mainly high acuity patients (i.e., ESI 1–2, and high risk ESI-3) under the supervision of Attendings in the main ED area. The study ED also provides Medical Student rotations year around. However, Medical Students were not allowed to formally document within ED patient medical records and Residents oversee Medical Student patients.

The study ED is continuously staffed with Attendings without any daily gaps in coverage. Residents are not scheduled during the 16 consecutive hours from Wednesday 2300 through Thursday 1500 each week to facilitate didactics. Attendings work solo during this time frame. Residents and Attendings are scheduled to work together during all other times weekly. Attendings are typically scheduled with one senior EM Resident (PGY-2 or PGY-3) and one junior Resident/off-service Resident (EM PGY-1 or non-EM). In general, teams are composed of one Attending and two Residents working within a fixed geographic location within the ED. Senior and junior residents are balanced when Attendings work with Residents. When Residents are not scheduled, these same geographic areas are staffed by solo Attendings.

This study divided patients into two groups. Attending working solo group included patients who were seen only by Attending physicians during the period Wednesday 2300 to Thursday 1500. Attending working with Residents group included patients who were seen by both the Attending and the Resident during all other times. However, if patients were only seen by Attending physicians during the Attending working with Residents time frame, these patients were included in the Attending working with Resident group. This was done because during that time frame, these Attendings have to supervise Residents simultaneously regardless of whether they are able to see patients by themselves. This occurs very rarely during Attending working with Resident shifts. EPIC® (Epic Systems Corporation, Verona, WI) electronic medical record (EMR) system was used for medical documentation.

### Participants

The study participants were EM Attendings and Residents. All patient encounters registered at the study ED and seen by participating Attendings and Residents during the period July 1, 2014 through June 30, 2017 were enrolled and analyzed. Patients who returned within 72 hours or were repeatedly seen at the study ED were considered as new patient encounters and treated as new patients for study purposes. As this study mainly focused on ED Attending performance, we excluded: 1) patients seen by other providers (e.g., APPs, non-EM Attendings);

2) part-time Attendings who worked fewer than 2 shifts per month; 3) Attendings who only worked solo or who only worked with Residents; and 4) Attendings who worked unbalanced shifts. An Attending with balanced shifts was defined as one who regularly worked both solo ($\geq$ 1 eight-hours shift per month) and with Residents ($\geq$ 4 eight-hours shifts per month) during the study period. Therefore, study Attendings were enrolled in both groups (Attending working solo versus Attending working with Residents) and we performed cross-over comparisons between these two groups.

## Data source

All data were retrieved from the EMR by persons from the hospital's Information Technology (IT) Department who were blinded to the study's outcomes. All data were subjected to internal validation assessment. Twenty random samples were selected at six separate phases and assessed manually by searching the EMR to determine the validity of the retrieved data.

## Variables

We collected patient general characteristics including age, gender, and ethnicity. Other variables included patient acuity level (ESI) at triage, patient total ED length of stay (LOS), provider-to-disposition time (PDT), the number of new patients per hour seen by a given provider, and ED crowding status [14] upon patient arrival to the ED. Detail variable explanations are addressed in S1.2 Appendix.

## Outcome measurements

We used three measurements: 1) the number of new patients per hour seen by a given Attending measured productivity; 2) PDT of each patient measured efficiency; and 3) Attending Performance Index (API) measured overall performance (see formula). API is a composite metric that integrates both productivity and efficiency. It is modified based on an established performance index [13]. A detail explanation of API is addressed in S1.3 Appendix.

$$Attending\ Performance\ Index = \frac{Number\ of\ New\ Patients\ per\ Hour\ per\ Attending}{(Acuity\ Level)^2 \times (Provider\ to\ Disposition\ Time\ in\ Hours)}$$

## Reporting guideline

We followed the STROBE (Strengthening the Reporting of Observational Studies in Epidemiology) reporting guideline in this study [15].

## Data analysis

We performed provider productivity, efficiency, and performance comparisons between Attendings working solo versus Attendings working with Residents at both the group and individual levels. We used Pearson chi square test for categorical data comparisons (gender, ethnicity, level of acuity, and ED crowding). For continuous data comparisons, we calculated mean with its standard deviation (SD) for the number of new patients per hour seen by a given provider and patient PDT. We also calculated median with its interquartile range (IQR) for age, the number of new patients per hour seen by a given provider, patient PDT, and API. A student $t$ test was used for mean comparisons between groups. A Wilcoxon rank-sum test was used for median comparisons between groups. To avoid potential confounders, 1:1 propensity score matching was performed between individual Attendings working solo and individual Attendings working with Residents after adjusting for patient age, gender, ethnicity, level of acuity, and ED crowding. To assess overall performance changes, the APIs of individual

Attendings were calculated and compared between individual Attendings working solo and working with Residents. Scant literature exists regarding overall performance measurement of Attendings and no quantitative benchmark value has been established to date indicating whether calculated API is predictive of Attending performance. Therefore, a threshold value API was determined in this study based on the changes of individual Attending APIs working with or without Residents. Indeed, this threshold value was a cutoff value at which different patterns of Attending overall performance were observed when working solo versus working with Residents. All analyses were performed using STATA® 14.2 (StataCorp LLC, College Station, TX) software with a p-value < 0.05 considered a statistically significant difference.

## Results

During the study period, a total of 49 ED Attendings and 271 ED Residents worked at the study ED. We excluded 14 part-time ED Attendings, 4 Attendings that either consistently worked solo or worked only with Residents, and 16 Attendings without balanced schedules. A final group of 15 ED Attendings and 266 Residents that collectively managed a total of 111,145 patient encounters during the study period was enrolled. (S1 Appendix Fig). Study patient general information is shown in Table 1. Patient and clinical characteristics (age, ESI, ED crowding, and ED LOS) in the final analysis differ between Attendings working solo and Attendings working with Residents groups (Table 1, p < 0.001). It was noted that the Attending working solo group saw a slightly more high-acuity (ESI 1) patients. More patients were

**Table 1. Study patient population general characteristics.**

| | Attendings Working Solo(N = 7,283) | Attendings Working with Residents(N = 103,871) | Total Patients Managed by Attendings Working Solo and Attendings Working with Residents Combined (N = 111,154) |
|---|---|---|---|
| Age—y (median, IQR)[a] | 48 (33, 58) | 47 (32, 57) | 47 (32, 57) |
| Gender—male (n, %)[b] | 3,855 (53) | 55,272 (53) | 59,127 (53) |
| Ethnicity—Hispanic (n, %)[c] | 1,802 (25) | 26,535 (26) | 28,337 (25) |
| ESI—(n,%)[a] | | | |
| ESI-1 | 294 (4.2) | 4,310 (4.0) | 4,604 (4.1) |
| ESI-2 | 2,391 (33) | 38,517 (37) | 40,908 (37) |
| ESI-3 | 3,953 (54) | 54,722 (53) | 58,675 (53) |
| ESI-4 | 531 (7.3) | 5,499 (5.3) | 6,030 (5.4) |
| ESI-5 | 99 (1.4) | 605 (0.6) | 704 (0.6) |
| Unknown | 15 (0.2) | 218 (0.2) | 233 (0.2) |
| ED Crowding—(n, %)[a] | | | |
| Not crowded | 3,949 (54) | 40,012 (39) | 43,961 (40) |
| Crowded | 1,669 (23) | 29,855 (29) | 31,524 (28) |
| Over-crowded | 1,665 (23) | 34,004 (33) | 35,669 (32) |
| ED Crowding—NEDOCS score (median, IQR)[a] | 92 (58, 136) | 115 (82, 154) | 114 (80, 153) |
| ED LOS—hours (median, IQR)[a] | 4.1 (2.7, 5.9) | 4.5 (3.1, 6.4) | 4.5 (3.1, 6.4) |

a:p < 0.001

b: p = 0.65

c: p = 0.19.

Abbreviations and definitions: IQR, Interquartile Range (25th, 75th); n, number; y, year; ED, Emergency Department; ESI, Emergency Severity Index; LOS, Length of Stay.

**Table 2. Provider productivity, efficiency, and performance measurements comparison between attendings working solo and attendings working with residents.**

| | Productivity Number of New Patients per Hour Median (IQR) Mean (SD) | Efficiency Provider to Disposition Time (Hours) Median (IQR) Mean (SD) | Performance* Attending Performance Index Median (IQR) |
|---|---|---|---|
| | Total Patients Before Propensity Score Matching (N = 111,154) | | |
| Attendings Working Solo | 3 (2, 4) [a] | 2.4 (1.4, 3.6) [c] | 0.16 (0.08, 0.36) [e] |
| | 2.9 (1.6) [a] | 2.7 (1.8) [c] | |
| Attendings Working with Residents | 3 (2, 4) | 2.7 (1.7, 3.9) | 0.17 (0.09, 0.34) |
| | 3.3 (1.9) | 3.0 (2.0) | |
| | Total Patients After Propensity Score Matching (N = 14,074) | | |
| Attendings Working Solo | 3 (2, 4) [b] | 2.4 (1.4, 3.6) [d] | 0.16 (0.08, 0.36) [f] |
| | 2.9 (1.6) [b] | 2.7 (1.8) [d] | |
| Attendings Working with Residents | 3 (2, 4) | 3.0 (1.9, 4.1) | 0.15 (0.08, 0.27) |
| | 3.1 (1.7) | 3.2 (1.9) | |

a: p < 0.001 (productivity comparison between Attendings Working Solo and Attendings Working with Residents)

b: p < 0.001 (productivity comparison between Attendings Working Solo and Attendings Working with Residents) using propensity score matching.

c: p < 0.001 (efficiency comparison between Attendings Working Solo and Attendings Working with Residents)

d: p < 0.001 (efficiency comparison between Attendings Working Solo and Attendings Working with Residents) using propensity score matching.

e: p = 0.037 (performance index comparison between Attendings Working Solo and Attendings Working with Residents)

f: p < 0.001 (performance index comparison between Attendings Working Solo and Attendings Working with Residents) using propensity score matching.

* Attending Performance Index (API) refers to formula: API = (number of new patients per hour seen by a provider)/((patient acuity level determined by ESI)$^2$ x (provider to disposition time in hours)).

Abbreviations and definitions: IQR, Interquartile Range; Provider to Disposition Time, time interval between initial provider encounter to disposition (i.e., admit, discharge, transfer) in hours

seen by the Attending working with Residents group than the Attending working solo group during times when the ED was overcrowded (Table 1, p < 0.001).

Analysis of productivity reveals more patients per hour were seen by Attendings working with Residents than Attendings working solo (Table 2). Total numbers of patients presenting to the ED from 2300 on a given day to 1500 the following day were calculated from Monday through Sunday revealing that patient volumes were within median range during the Attending working solo timeframe (Wednesday 2300 to Thursday 1500) (S2 Appendix). Shorter PDT (i.e., efficiency) is noted in the Attending working solo group (Table 2). Essentially, Attending productivity increased but efficiency decreased when working with Residents. Overall performance seems to be increased among Attendings working with Residents in comparison to Attendings working solo. However, the benefit of increased Attending provider performance while working with Residents appears to be diminished when propensity score matching is applied (Table 2).

Additionally, when Attending productivity, efficiency, and overall performance are measured at an individual Attending level, the case of increased productivity with decreased efficiency is observed for most individual Attendings when working with Residents compared to working solo (S3 Appendix and S4 Appendix). All 15 Attendings were divided into two groups in terms of their solo performance. Higher Attending performance indexes (APIs) are observed among individual Attendings when working with Residents if their solo API < 0.18. On the contrary, lower APIs were noted among individual Attendings whose solo API ≥ 0.18

**Table 3. Attending overall performance comparisons between attendings working solo versus attendings working with residents based on attending solo performance.**

| | Original Data | | | Propensity Score Matching Data | | |
|---|---|---|---|---|---|---|
| | Attendings Working Solo Median (IQR) | Attendings Working with Residents Median (IQR) | p | Attendings Working Solo Median (IQR) | Attendings Working with Residents Median (IQR) | p |
| Productivity | | | | | | |
| Attendings with low baseline API | 2 (2, 3) | 3 (2, 4) | < 0.001 | 2 (2, 3) | 3 (2, 4) | < 0.001 |
| Attendings with high baseline API | 3 (2, 4) | 3 (2, 5) | < 0.001 | 3 (2, 4) | 3 (2, 4) | 0.009 |
| Efficiency | | | | | | |
| Attendings with low baseline API | 2.8 (1.8, 4.0) | 2.8 (1.8, 4.1) | 0.018 | 2.8 (1.8, 4.0) | 3.0 (2.0, 4.3) | < 0.001 |
| Attendings with high baseline API | 2.1 (1.1, 3.2) | 2.6 (1.6, 3.8) | < 0.001 | 2.1 (1.1, 3.2) | 2.8 (1.8, 4.0) | < 0.001 |
| Performance index | | | | | | |
| Attendings with low baseline API | 0.13 (0.07, 0.26) | 0.16 (0.08, 0.31) | < 0.001 | 0.13 (0.07, 0.26) | 0.14 (0.07, 0.25) | < 0.001 |
| Attendings with high baseline API | 0.21 (0.11, 0.48) | 0.19 (0.05, 0.38) | < 0.001 | 0.21 (0.11, 0.48) | 0.15 (0.08, 0.29) | < 0.001 |

Abbreviations and definitions: API, Attending Performance Index; IQR, Interquartile Range; Performance measure = Attending Performance Index [(number of new patients per hour seen by a provider)/((patient acuity level determined by ESI)$^2$ x (provider to disposition time in hours))].

when working with Residents (S5 Appendix). This pattern remained when confounding factors were adjusted using the 1:1 propensity score matching model (Table 3). Therefore, a threshold API (high API ≥ 0.18 and low API < 0.18) of individual Attending performance was determined in this study.

## Discussion

Our study found increased productivity with decreased efficiency among Attendings working with Residents. When API was used for performance assessment, we found that individual Attendings with high solo performances rendered an overall decreased performance while working with Residents. On the contrary, working with Residents increased overall performance among Attendings with low solo API. Though the study was performed in an ED, interpretation of study findings might not be limited to the ED setting since they reflect a similar academic teaching model whereby Attending physicians work with Residents in most clinical practice settings. Therefore, our results add more evidence to summative provider performance measurements in an academic institution thereby providing valuable insight to current and future approaches to Resident training and faculty evaluation across different residency specialty programs.

Productivity and efficiency can be affected by differences in patient acuity. The study ED preferentially flows patients such that Attendings working solo and Attendings working with Residents provide high acuity care and the APP staff provide low acuity care. Though the majority of high acuity patients (ESI 1–2) were seen by Attendings regardless of whether they worked solo or with Residents, productivity and efficiency differences occurred between these two groups. The possible reasons might be: 1) the daily variety of high acuity patients present at the study ED (e.g. slightly more ESI-1 patients present during non-resident shifts); and 2) the exclusion of the APPs in this study. During the ED non-resident time (16h/week), an extra

APP was scheduled to work. However, this study excluded APPs who usually only saw low-acuity patients. Under this circumstance, it is necessary to use propensity score matching to minimize potential confounder effects in this study. When propensity score matching was applied, the median number of new patients per hour seen by either Attendings working solo or Attendings working with Residents reveals no changes both at the group (Table 2) and individual (Table 3) levels, indicating that patient acuity level has no effect on provider productivity measurement. In addition, different metrics have been used to measure provider productivity in the literature, including RVUs or number of new patients per shift. However, RVUs can be significantly affected based on encounter documentation alone. Our providers usually spend extra time outside of their clinical shift to complete documentation, thus it is extremely difficult and inaccurate to add extra time for documentation when deriving the provider productivity calculation. Moreover, providers working at the study ED typically work different length shifts (e.g., 8h, 9h, 10h, or 12h shifts), serving as a potential confounder that fails to compare provider productivity on a common basis when number of new patients per *shift* is used. Therefore, to simplify this study, the number of new patients per *hour*, as opposed to RVUs per hour and/or patients per *shift*, was used as our Attending productivity measurement [4,16]. On the other hand, we used PDT instead of patient LOS for provider efficiency measurement because LOS is often affected by different system- and patient-associated variables, which could be potential confounders in the context of provider efficiency (e.g., ED crowding, waiting room time, etc.) [17,18]. Unlike LOS, PDT is more directly affected by a given provider during the decision-making process impacting downstream diagnostic and therapeutic resource needs turnaround intervals [4,13]. Therefore, PDT delivers better interpretive quality regarding Attending efficiency. Ideally, a higher number of new patients per hour (high productivity) along with a lower PDT per patient (high efficiency) is considered the most desirable overall provider performance outcome combination. A balanced provider performance index was therefore used in this study [13].

We refrained from including all ED Attending data in this analysis to avoid developing different biases. We believe that inclusion of ED Attendings with non-balanced shift schedules based either on working solo or working with Residents arms will produce bias thereby complicating comparison of performance changes at an individual level. Furthermore, provider productivity and efficiency could vary significantly based on provider proficiency levels, thereby potentially affecting Attending productivity and efficiency when working non-balanced shifts. Working non-balanced shifts will affect Attending proficiency when spending more time performing unusual tasks (e.g., increased documentation time in absence of Resident input to overall encounter workflow). We therefore only enrolled Attendings with balanced shift schedules and analyzed median values to stabilize potential confounders. As shown in Tables 2 and 3, no changes to median values are noted after all potential confounders are adjusted by propensity score matching. Such findings further confirm the stability of using medians (IQR) for productivity and performance measurements.

Many previous studies investigated the impact of trainees on ED Attending performance concluding diverse findings [7,11,19,20]. When patient ED LOS was measured, some studies showed that ED LOS was not affected significantly by the presence of trainees rotating in the ED [11], whereas other studies showed an increase in ED LOS when Attendings worked with Residents [19,20]. In one study, an average of 7 minutes of increased LOS was reported with each additional trainee working with a given Attending [20]. In terms of provider productivity, a previous study also revealed no significant difference in number of patients per hour seen by junior EM, senior EM, or off-service Residents [7]. The reasons that our study findings differ from previous reports include: 1) we balanced our samples of Attendings in both groups, therefore the changes in their operational performance can be compared at an individual

Attending level. In contrast, other studies mainly assess the operational performance at a group level; 2) we used PDT instead of LOS for provider efficiency measurement which may help reduce the effects of other confounding factors (e.g., ED crowding). The use of propensity score matching further minimizes potential biases; 3) the study ED had a 1:2 Attending-to-Resident ratio instead of 1:1 ratio staffing. This might affect Attending performance more significantly than the 1:1 teaching model, which could indirectly result in increased Attending productivity (i.e., more providers are available to see patients during the same time interval) with decreased efficiency (i.e., more trainees requiring more supervision per Attending per unit time), which differs from the previous report [7].

Our study has its limitations. First, this is a single center retrospective study which cannot demonstrate causality due to potentially harvesting incorrect information and selection bias. Second, although two key performance metrics were used for provider productivity and efficiency analysis, other metrics (e.g., number of RVUs per hour per provider, number of patients treated per shift) were not collected and compared in this study. Additionally, productivity and efficiency could be affected multi-factorially to include patient disease severity, nursing staffing levels, efficiency levels of nursing staff, etc. Study results might be inaccurate without the measure of individual patient disease severity and nursing staffing and their efficiencies. Third, among the Attending working with Residents group, due to differences in Resident efficiency and performance, Attending productivity, efficiency, and performance might potentially be affected. We also did not analyze and interpret how different levels of resident (EM versus non-EM residents) and medical students could impact attendings' productivity, efficiency, and performance. However, we consider such effects to be minimal due to 1) blinded scheduling of Attendings and Residents resulting in random Attending-Resident shift combinations thereby avoiding significant heterogeneity; and 2) this is a 3-year study with over 100,000 patients seen by 15 Attendings working with 266 different Residents therefore each Attending had opportunity to work with Residents performing at different efficiency levels producing minimal overall effect on performance. Fourth, under rare conditions, Attendings might see patients by themselves while also supervising Residents during the Attending working with Residents shifts. Though it rarely occurs (< 1% in this study), it could still potentially affect our study results. Last, patients in the Attending working solo group accounted for a relatively small portion of the entire study population due to the current staffing model that employs Residents 90% of the available weekly schedule (152/168 hours). Given the fact of a fixed schedule of Attending only shifts, patient volume differences and potential patient selection bias could possibly occur when compared to patients from other groups. Therefore, a multi-center prospective study is warranted for external validation.

## Conclusion

Diverse Attending productivity and efficiency exists. In our study, Attending overall performance, when working with Residents, varied inversely when compared to their performance working solo.

## Supporting information

**S1 Appendix. (S1.1 Appendix: Detail Explanation of Emergency Severity Index (ESI); S1.2 Appendix: Detail Variables Explanation; S1.3 Appendix: Detail Explanation of Attending Performance Index (API)).**
(DOCX)

**S2 Appendix. Daily patient volume as measured during time interval 2300 (previous day) through 1500 (next day).**
(DOCX)

**S3 Appendix. Productivity comparisons between attendings working solo versus attendings working with residents.**
(DOCX)

**S4 Appendix. Efficiency (provider to disposition time) comparisons between attendings working solo versus attendings working with residents.**
(DOCX)

**S5 Appendix. Overall performance comparisons between attendings working solo versus attendings working with residents.**
(DOCX)

**S1 Appendix Fig. Study flow diagram.**
(PDF)

## Author Contributions

**Conceptualization:** Hao Wang.

**Data curation:** Sasha Dib, Daisha Mclarty, Sajid Shaikh, Radhika Cheeti, Hao Wang.

**Formal analysis:** Yuan Zhou, Yasaman Ghasemi, Mdmamunur Rahman, Hao Wang.

**Investigation:** Richard D. Robinson, Sasha Dib, Daisha Mclarty, Sajid Shaikh, Radhika Cheeti, Hao Wang.

**Methodology:** Richard D. Robinson, Hao Wang.

**Project administration:** Hao Wang.

**Supervision:** Richard D. Robinson, Chet D. Schrader, Hao Wang.

**Validation:** Richard D. Robinson, Sasha Dib, Daisha Mclarty, Sajid Shaikh, Radhika Cheeti, Hao Wang.

**Writing – original draft:** Richard D. Robinson, Hao Wang.

**Writing – review & editing:** Richard D. Robinson, Sasha Dib, Daisha Mclarty, Sajid Shaikh, Radhika Cheeti, Yuan Zhou, Yasaman Ghasemi, Mdmamunur Rahman, Chet D. Schrader, Hao Wang.

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
