## [Decision Letter · Decision Letter 0]

12 Nov 2019

PONE-D-19-26595

Productivity, Efficiency, and Overall Performance Comparisons Between Solo Attending Versus Attending with Residents Staffing Models in an Emergency Department

PLOS ONE

Dear Dr. Wang,

Thank you for submitting your manuscript to PLOS ONE. After careful consideration, we feel that it has merit but does not fully meet PLOS ONE’s publication criteria as it currently stands. Therefore, we invite you to submit a revised version of the manuscript that addresses the points raised during the review process.

We would appreciate receiving your revised manuscript by Dec 27 2019 11:59PM. To enhance the reproducibility of your results, we recommend that if applicable you deposit your laboratory protocols in protocols.io, where a protocol can be assigned its own identifier (DOI) such that it can be cited independently in the future. For instructions see: http://journals.plos.org/plosone/s/submission-guidelines#loc-laboratory-protocols

We look forward to receiving your revised manuscript.

Kind regards,

Andrew Carl Miller

Academic Editor

PLOS ONE

Journal Requirements:

1. We note that you have indicated that data from this study are available upon request. PLOS only allows data to be available upon request if there are legal or ethical restrictions on sharing data publicly. For information on unacceptable data access restrictions, please see http://journals.plos.org/plosone/s/data-availability#loc-unacceptable-data-access-restrictions.

Reviewers' comments:

Reviewer's Responses to Questions

**Comments to the Author**

1. Is the manuscript technically sound, and do the data support the conclusions?

Reviewer #1: No

Reviewer #2: Yes

2. Has the statistical analysis been performed appropriately and rigorously? 

Reviewer #1: I Don't Know

Reviewer #2: Yes

3. Have the authors made all data underlying the findings in their manuscript fully available?

Reviewer #1: Yes

Reviewer #2: Yes

4. Is the manuscript presented in an intelligible fashion and written in standard English?

Reviewer #1: No

Reviewer #2: Yes

5. Review Comments to the Author

Reviewer #1: -Based on the data presented in the results section of the abstract, the conclusion that “increased productivity […] often occurs among attendings working with residents” isn’t supported by the data cited as you cited identical median numbers. If other data shows this, cite that instead. Also I would take out the word “often.”

First paragraph: However, both are affected by multiple factors including ED crowding, patient acuity, (AND) supervision of Residents and Medical Students. [5-7].

Second paragraph: “Whereas, another study determined that ED LOS was not significantly affected by the presence or total number of trainees in the ED [11]” is a sentence fragment. Change to “However, another study determined that ED LOS was not significantly affected by the presence or total number of trainees in the ED [11].”

Second paragraph: “At present, few studies compare provider productivity and efficiency between Attendings working solo versus Attendings working with Residents.” How does this study differ from or add to what we already have in the literature on this?

Third paragraph: “Overall provider performance may be gained (change to “calculated” or “defined”)by combining productivity and efficiency arriving at a composite measure [13].” This paragraph at the end of the introduction does not make sense all by itself. If you want to introduce a new metric, I would do it in the first paragraph where you start defining the metrics used.

The paper switches frequently between passive and active tense; this needs to be more consistent.

In the results section, the conclusion that “Analysis of productivity reveals more patients per hour were seen by Attendings working with Residents than Attendings working solo (Table 2).” Is not supported by the data cited which shows identical median patient numbers for both groups.

In the results section, why was the cut off of 0.18 used for the API for high index or low index categorization? Has this been looked at and defined in the past?

Discussion: First sentence “Our study found increased productivity with decreased efficiency among Attendings working with Residents.” Is not supported by the data presented which cites identical median productivity numbers. How are you getting to this conclusion?

“Though the majority of high acuity patients (ESI 1-2-3) were seen by Attendings regardless of whether they worked solo or with Residents, differences occurred between these two groups.” What differences are these?

Discussion:

“Therefore, to simplify this study, the number of new patients per hour as opposed to RVUs per hour was used as our Attending productivity measurement [4,15]. On the other hand, we used PDT instead of patient LOS for provider efficiency measurements because LOS is often affected by different system and patient associated variables serving as meaningful confounders (e.g., ED crowding, waiting room time, etc.) [16,17]. Unlike LOS, PDT is more directly affected by a given provider during the decision-making process and resultant downstream diagnostic and therapeutic resource needs [4,13]. Therefore, PDT delivers better interpretive quality regarding Attending efficiency.” All of this likely would do better in the methods section as it does not discuss the results of your study.

Reviewer #2: General:

- Inserting line numbers will make it easier for reviewers to provide more focused feedback.

- Please be sure to list the appropriate guideline used and provide citation. For example:

von Elm E, Altman DG, Egger M, Pocock SJ, Gotzsche PC, Vandenbroucke JP. The Strengthening the Reporting of Observational Studies in Epidemiology (STROBE) Statement: guidelines for reporting observational studies. PLoS Med. 2007;4(10):e296. PMID: 17941714

Title:

- Indicate the study’s design with a commonly used term in the title or the abstract

Abstract:

- Please reposition the headings Methods, Results, and Conclusions so that they are not contiguous with the preceding paragraph.

Introduction:

- Pg 4: Resident efficiency has been reported as less than that of APPs. How does this compare with the number of bounce-back or repeat visits? The number of M&M cases? APP practice environments vary greatly; are the APPs in this comparison practicing autonomously (attending consult available but not required), or do they have to present their cases to an attending still? Were they APPs who had done an EM “residency” or practicing straight out of school?

- Same line as above: is there any evidence that the APPs document better, thereby billing at a higher level and generating more RVUs? To aid discussion of these points, one my consider incorporating the reference by McDonnell into the discussion (PMID: 25654675 DOI: 10.1097/PEC.0000000000000349).

Methods:

- The ESI is a poor judge of illness severity. From the EMR one may be able to obtain the necessary info to determine illness severity using a validated tool such as Charlson Comorbidity Index, or 3M-APR-DRG. This would be helpful for inter-group comparison.

- Explain how the study size was arrived at. Provide a calculation to justify sample size & method used.

- STATA 14.2 software. It is convention to list the manufacturer & location in parentheses after the name. Additional ™ or ® should be listed if applicable.

- Do the physicians chart using scribes (see PMID: 30700408; PMID: 27856140), dictation, manual electronic, or paper charting?

- Was the presence of medical students and other learners recorded and factored in?

- Were the same attendings enrolled in both groups depending on whether or not they had a resident, or was there no cross-over?

- Just to confirm, were all residents emergency medicine residents, or were some off-service residents (eg. internal medicine, ob-gyn, etc.)?

Results

- Was time of shift and staffing levels recorded? Did these differ between groups? If so, changes in nursing staffing could also have impacted the results.

- What was the distribution of the year of resident training (1, 2, 3, etc.)? It would be important to know if it was not balanced between senior and junior residents.

Discussion

- Would benefit from a deeper discussion of why they think they found no difference when others have. Compare and contrast with the results of other published studies on the topic including: (PMID: 24578767; PMID: 24672605; PMID: 25972206; PMID: 24238313; PMID: 18973640)

6. PLOS authors have the option to publish the peer review history of their article (what does this mean?). If published, this will include your full peer review and any attached files.

Reviewer #1: Yes: Marina Boushra

Reviewer #2: No

---

## [Author Response · Author response to Decision Letter 0]

16 Dec 2019

AUTHORS RESPONSE TO EDITORS AND REVIEWERS

As requested, we have included the original letter and comments with our point by point response in red colored font.

PONE-D-19-26595

Productivity, Efficiency, and Overall Performance Comparisons Between Solo Attending Versus Attending with Residents Staffing Models in an Emergency Department

PLOS ONE

Dear Dr. Wang,

Thank you for submitting your manuscript to PLOS ONE. After careful consideration, we feel that it has merit but does not fully meet PLOS ONE’s publication criteria as it currently stands. Therefore, we invite you to submit a revised version of the manuscript that addresses the points raised during the review process.

We would appreciate receiving your revised manuscript by Dec 27 2019 11:59PM. To enhance the reproducibility of your results, we recommend that if applicable you deposit your laboratory protocols in protocols.io, where a protocol can be assigned its own identifier (DOI) such that it can be cited independently in the future. For instructions see: http://journals.plos.org/plosone/s/submission-guidelines#loc-laboratory-protocols

• A rebuttal letter that responds to each point raised by the academic editor and reviewer(s). This letter should be uploaded as separate file and labeled 'Response to Reviewers'. 

• A marked-up copy of your manuscript that highlights changes made to the original version. This file should be uploaded as separate file and labeled 'Revised Manuscript with Track Changes'. 

• An unmarked version of your revised paper without tracked changes. This file should be uploaded as separate file and labeled 'Manuscript'.

We look forward to receiving your revised manuscript.

Kind regards,

Andrew Carl Miller

Academic Editor

PLOS ONE

Journal Requirements:

1. We note that you have indicated that data from this study are available upon request. PLOS only allows data to be available upon request if there are legal or ethical restrictions on sharing data publicly. For information on unacceptable data access restrictions, please see http://journals.plos.org/plosone/s/data-availability#loc-unacceptable-data-access-restrictions.

Response: Yes, our data contain identifiable patient information including patient name, age, and admission date/time. This project was approved by the local Institutional Review Board (IRB). We provided information of the contact person to whom data requests may be sent.

Response: Yes, we revised and included captions for our supplemental figure at the end of our manuscript.

Reviewers' comments:

Reviewer's Responses to Questions

Comments to the Author

1. Is the manuscript technically sound, and do the data support the conclusions?

Reviewer #1: No

Reviewer #2: Yes

2. Has the statistical analysis been performed appropriately and rigorously? 

Reviewer #1: I Don't Know

Reviewer #2: Yes

3. Have the authors made all data underlying the findings in their manuscript fully available?

Reviewer #1: Yes

Reviewer #2: Yes

4. Is the manuscript presented in an intelligible fashion and written in standard English?

Reviewer #1: No

Reviewer #2: Yes

5. Review Comments to the Author

Reviewer #1: 

1. -Based on the data presented in the results section of the abstract, the conclusion that “increased productivity […] often occurs among attendings working with residents” isn’t supported by the data cited as you cited identical median numbers. If other data shows this, cite that instead. Also I would take out the word “often.”

Response: Although no obvious MEDIAN differences were observed between Attendings working solo versus working with Residents by reviewing the median (IQR) of the performance measures, the Wilcoxon-Rank Sum test did show statistically significant differences in their means (p-value <0.001). Wilcoxon-Rank Sum test is a non-parametric statistical test for comparing means of two samples that may not be normally distributed. Sorry for the confusion. In order to better interpret our data, we revised and added the mean (SD) along with the median (IQR) in Table 2 (statistically significant differences between these two groups were found). We used mean (SD) of productivity and efficiency reported in the abstract (Results section). In addition, we deleted “often” under the Conclusions section of the abstract. 

2. First paragraph: However, both are affected by multiple factors including ED crowding, patient acuity, (AND) supervision of Residents and Medical Students. [5-7].

Second paragraph: “Whereas, another study determined that ED LOS was not significantly affected by the presence or total number of trainees in the ED [11]” is a sentence fragment. Change to “However, another study determined that ED LOS was not significantly affected by the presence or total number of trainees in the ED [11].”

Response: Yes, we revised and added “and” between “patient acuity,” and “supervision of Residents and Medical Students.”. We also revised, and changed to “However, another study determined that ED LOS was not significantly affected by the presence of total number of trainees in the ED [11].” (see introduction section, the first and second paragraphs).

3. Second paragraph: “At present, few studies compare provider productivity and efficiency between Attendings working solo versus Attendings working with Residents.” How does this study differ from or add to what we already have in the literature on this?

Response: Thanks for reviewer’s valued comment. We revised and added the importance of this study to address how it differs from other studies reported in the literature (see below). It was inserted in the second paragraph of the Introduction section.

 “At present, only a few studies compare provider productivity and efficiency between Attendings working solo versus Attendings working with Residents, and none of them examine the differences in productivity and efficiency of individual Attendings within these two groups.”

4. Third paragraph: “Overall provider performance may be gained (change to “calculated” or “defined”)by combining productivity and efficiency arriving at a composite measure [13].” This paragraph at the end of the introduction does not make sense all by itself. If you want to introduce a new metric, I would do it in the first paragraph where you start defining the metrics used.

Response: Yes, we revised and changed “gained” to “calculated”. We realize this was confusing. Sorry about that. The whole paragraph mentioned above is indeed part of the literature review. We intend to report a composite index was developed previously to measure providers’ overall performance; however, this index lacks external validation. In terms of the flow of the paper, we think it would be appropriate to describe it here together with other existing work. 

”Productivity and efficiency are often consistent when used to evaluate provider operational performance (i.e., efficient providers are also productive or vice versa). However, it is not uncommon to observe some inconsistencies (e.g., high productivity but low efficiency), thus creating a significant challenge to understanding provider overall operational performance. To overcome this challenge, prior research introduced a composite index, which was calculated by combining productivity and efficiency [13]. However, a major shortcoming of said calculation is the lack of external validation.”

5. The paper switches frequently between passive and active tense; this needs to be more consistent.

Response: Yes, we revised the entire manuscript to be more consistent.

6. In the results section, the conclusion that “Analysis of productivity reveals more patients per hour were seen by Attendings working with Residents than Attendings working solo (Table 2).” Is not supported by the data cited which shows identical median patient numbers for both groups.

Response: Again, sorry for the confusion. We revised it and added both median (IQR) and mean (SD) of productivity and efficiency of attendings solo versus attendings working with residents (see revised Table 2).

7. In the results section, why was the cut off of 0.18 used for the API for high index or low index categorization? Has this been looked at and defined in the past?

Response: No, the cutoff of 0.18 has not been reported in the past. This number was derived by interpreting this study’s data at the individual provider level (see Append Table 5). We revised and added explanations in the Methods and Results sections as follows:

1. In the Methods section (last paragraph of the Methods section: data analysis) 

“To assess overall performance changes, the APIs of individual Attendings were calculated and compared between individual Attendings working solo and working with Residents. Scant literature exists regarding overall performance measurement of Attendings and no quantitative benchmark value has been established to date indicating whether calculated API is predictive of Attending performance. Therefore, a threshold value API was determined in this study based on the changes of individual Attending APIs working with or without Residents. Indeed, this threshold value was a cutoff value at which different patterns of Attending overall performance were observed when working solo and working with Residents.”

2. In the Results section (third paragraph of the Results section)

“Additionally, when Attending productivity, efficiency, and overall performance are measured at an individual Attending level, the case of increased productivity with decreased efficiency is observed for most individual Attendings when working with Residents compared to working solo (Appendix-3 and Appendix-4). All 15 Attendings are divided into two groups in terms of their solo performance. Higher Attending performance indexes (APIs) are observed among individual Attendings when working with Residents if their solo API < 0.18. On the contrary, lower APIs were noted among individual Attendings whose solo API ≥ 0.18 when working with Residents (Appendix-5). This pattern remained when confounding factors were adjusted using the 1:1 propensity score matching model (Table 3). Therefore, a threshold API (high API ≥ 0.18 and low API < 0.18) of individual Attending performance was determined in this study.”

8. Discussion: First sentence “Our study found increased productivity with decreased efficiency among Attendings working with Residents.” Is not supported by the data presented which cites identical median productivity numbers. How are you getting to this conclusion?

Response: As we addressed before, we added our mean (SD) of productivity and efficiency to both attendings working solo and attendings working with resident groups. Both showed statistically significant differences. (see response of comment #1 and Table 2)

9. “Though the majority of high acuity patients (ESI 1-2-3) were seen by Attendings regardless of whether they worked solo or with Residents, differences occurred between these two groups.” What differences are these?

Response: Sorry for the confusion. We revised this sentence and added “productivity and efficiency differences”. Please see the revised following:

“Though the majority of high acuity patients (ESI 1-2-3) were seen by Attendings regardless of whether they worked solo or with Residents, productivity and efficiency differences occurred between these two groups.”

10. Discussion:

“Therefore, to simplify this study, the number of new patients per hour as opposed to RVUs per hour was used as our Attending productivity measurement [4,15]. On the other hand, we used PDT instead of patient LOS for provider efficiency measurements because LOS is often affected by different system and patient associated variables serving as meaningful confounders (e.g., ED crowding, waiting room time, etc.) [16,17]. Unlike LOS, PDT is more directly affected by a given provider during the decision-making process and resultant downstream diagnostic and therapeutic resource needs [4,13]. Therefore, PDT delivers better interpretive quality regarding Attending efficiency.” All of this likely would do better in the methods section as it does not discuss the results of your study.

Response: Thanks for reviewer’s valued comment. We considered it necessary to discuss the reason we use the number of new patients per hour as a parameter for provider productivity measurement in this study instead of using others (e.g. RVU, number of new patients per shift). Same as efficiency measurement, we addressed why we use PDT instead of patient LOS as the parameter for efficiency measurement. We believe a discussion on the variety of operational metrics used for productivity/efficiency measurements might be suitably placed in the Discussion section. However, we understand the confusion, we revised our discussion as follows (see the second paragraph under Discussion section).

“Productivity and efficiency can be affected by differences in patient acuity. The study ED preferentially flows patients such that Attendings working solo and Attendings working with Residents provide high acuity care and the APP staff provide low acuity care. Though the majority of high acuity patients (ESI 1-2-3) were seen by Attendings regardless of whether they worked solo or with Residents, productivity and efficiency differences occurred between these two groups. Therefore, it is necessary to use propensity matching to minimize potential confounder effects in this study. When propensity score matching was applied, the median number of new patients per hour seen by either Attendings working solo or Attendings working with Residents reveals no changes both at the group (Table 2) and individual (Table 3) levels, indicating that patient acuity level has no effect on provider productivity measurement. In addition, different metrics have been used to measure provider productivity in the literature, including RVUs or number of new patients per shift. However, RVUs can be significantly affected based on encounter documentation alone. Our providers usually spend extra time outside of their clinical shift to complete documentation, thus it is extremely difficult and inaccurate to add extra time for documentation when deriving the provider productivity calculation. Moreover, providers working at the study ED typically work different length shifts (e.g., 8h, 9h, 10h, or 12h shifts), serving as a potential confounder that fails to compare provider productivity on a common basis when number of new patients per shift is used. Therefore, to simplify this study, the number of new patients per hour, as opposed to RVUs per hour and/or patients per shift, was used as our Attending productivity measurement [4,15]. On the other hand, we used PDT instead of patient LOS for provider efficiency measurement because LOS is often affected by different system- and patient-associated variables, which could be potential confounders in the context of provider efficiency (e.g., ED crowding, waiting room time, etc.) [16,17]. Unlike LOS, PDT is more directly affected by a given provider during the decision-making process impacting downstream diagnostic and therapeutic resource needs turnaround intervals [4,13]. Therefore, PDT delivers better interpretive quality regarding Attending efficiency. Ideally, a higher number of new patients per hour (high productivity) along with a lower PDT per patient (high efficiency) is considered the most desirable overall provider performance outcome combination. A balanced provider performance index was therefore used in this study [13]. ”

Reviewer #2: General:

11. - Inserting line numbers will make it easier for reviewers to provide more focused feedback.

Response: yes, we inserted line numbers to the manuscript.

12. - Please be sure to list the appropriate guideline used and provide citation. For example:

von Elm E, Altman DG, Egger M, Pocock SJ, Gotzsche PC, Vandenbroucke JP. The Strengthening the Reporting of Observational Studies in Epidemiology (STROBE) Statement: guidelines for reporting observational studies. PLoS Med. 2007;4(10):e296. PMID: 17941714

Response: Yes, we revised and added the appropriate guideline under the method section. (see method section, under reporting guideline)

13. Title:

- Indicate the study’s design with a commonly used term in the title or the abstract

Response: Yes, we revised the title as the following:

“Productivity, Efficiency, and Overall Performance Comparisons Between Solo Attending Versus Attending with Residents Staffing Models in an Emergency Department: A Large-scale Retrospective Observational Study”

14. Abstract:

- Please reposition the headings Methods, Results, and Conclusions so that they are not contiguous with the preceding paragraph.

Response: Yes, they are revised.

15. Introduction:

- Pg 4: Resident efficiency has been reported as less than that of APPs. How does this compare with the number of bounce-back or repeat visits? The number of M&M cases? APP practice environments vary greatly; are the APPs in this comparison practicing autonomously (attending consult available but not required), or do they have to present their cases to an attending still? Were they APPs who had done an EM “residency” or practicing straight out of school?

Response: Thanks for reviewer’s valued comment. Patients who had bounce-back or repeated visits will be counted as another patient encounter and treated as a new patient. In this study, we did not specifically address the number of M&M cases. We understand APP practice differs in different environments. Therefore, in this study, we excluded patients seen by APPs. In the study ED, APPs see patients independently and can disposition patients by themselves without the need to notify attending physicians. Under very rare circumstances, an APP might call an attending physician to get a second opinion on their patients since APPs only see low acuity patients. All APPs will need ED experience before they work at the study ED. We realize such uncertainty, we revised and addressed these under the Methods section. 

16. - Same line as above: is there any evidence that the APPs document better, thereby billing at a higher level and generating more RVUs? To aid discussion of these points, one my consider incorporating the reference by McDonnell into the discussion (PMID: 25654675 DOI: 10.1097/PEC.0000000000000349).

Response: Since this project mainly discussed attending and resident productivity, efficiency and their performance, we did not include APPs in this study. 

17. Methods:

- The ESI is a poor judge of illness severity. From the EMR one may be able to obtain the necessary info to determine illness severity using a validated tool such as Charlson Comorbidity Index, or 3M-APR-DRG. This would be helpful for inter-group comparison.

Response: Thanks for reviewer’s valued comment. We understand that ESI is a poor indicator for illness severity. Unfortunately, due to the nature of this study’s design, we are unable to add CCI or APR-DRG to the final analysis. We revised and addressed these in our limitation section. (see the fifth paragraph under discussion section)

18. - Explain how the study size was arrived at. Provide a calculation to justify sample size & method used.

Response: This is a retrospective study and our intent is to include all patients seen by attendings and residents during the study period. Therefore, we did not perform the sample size estimation in this study.

19. - STATA 14.2 software. It is convention to list the manufacturer & location in parentheses after the name. Additional ™ or ® should be listed if applicable.

Response: Yes, it is revised and added. (see last paragraph under method section)

20. - Do the physicians chart using scribes (see PMID: 30700408; PMID: 27856140), dictation, manual electronic, or paper charting?

Response: EMR (electronic medical record) system was used for provider charting in this study. We revised and added under the method section. 

21. - Was the presence of medical students and other learners recorded and factored in?

Response: Yes, our ED provides medical students rotation all year around. However, medical students are not allowed to document under EMR system. Therefore, residents oversee medical students’ patients. We revised and addressed it under the method section.

22. - Were the same attendings enrolled in both groups depending on whether or not they had a resident, or was there no cross-over?

Response: Yes, the same attendings were enrolled in both groups and this is a cross-over comparison study. We revised and added to the method section.

23. - Just to confirm, were all residents emergency medicine residents, or were some off-service residents (eg. internal medicine, ob-gyn, etc.)?

Response: Yes, study included off-service residents (non-EM residents). These non-EM residents were treated as PGY-1 EM residents. We addressed it under the method section.

24. Results

- Was time of shift and staffing levels recorded? Did these differ between groups? If so, changes in nursing staffing could also have impacted the results.

Response: Yes, time of the shift was recorded, however, due to the different length of shifts (e.g. 8h, 9h, 10h, 12h-shifts), this makes it hard to calculate number of patients per shift. Therefore, we use number of new patients per HOUR for productivity measurement. We understand that nursing staff will also affect productivity and efficiency, unfortunately, we did not record nursing staffing levels nor record the experience levels of nursing staff (RN1 versus RN2 versus more advanced RN levels). We revised and addressed it in the limitation section under the discussion. (see fifth paragraph under discussion section)

25. - What was the distribution of the year of resident training (1, 2, 3, etc.)? It would be important to know if it was not balanced between senior and junior residents.

Response: We have balanced distribution of residents with different levels of training. Each month, we have approximately equal numbers of senior EM residents, PGY-1 EM residents, and non-EM residents. We revised under the method section as the following:

“Attendings are typically scheduled with one senior EM Resident (PGY-2 or PGY-3) and one junior Resident (EM PGY-1 or non-EM). In general, teams are composed of one Attending and two Residents working within a fixed geographic location within the ED. Senior and junior residents are balanced when Attendings working with the residents.” 

26. Discussion

- Would benefit from a deeper discussion of why they think they found no difference when others have. Compare and contrast with the results of other published studies on the topic including: (PMID: 24578767; PMID: 24672605; PMID: 25972206; PMID: 24238313; PMID: 18973640)

Response: Thanks for reviewer’s valued comments. We reviewed all mentioned references, some studies have been cited in this manuscript (PMID 24578767, PMID25972206), while others have not. We realize that our study required deeper discussion. Therefore, we revised and added a separate paragraph under the Discussion section to further discuss the differences when attending physicians work with trainees. See the following revision (the fourth paragraph under discussion section).

“Many previous studies investigated the impact of trainees on ED Attending performance concluding diverse findings [7,11,18,19]. When patient ED LOS was measured, some studies showed that ED LOS was not affected significantly by the presence of trainees rotating in the ED [11], whereas other studies showed an increase in ED LOS when Attendings worked with Residents [18,19]. An average of 7 minutes of increased LOS was reported with each additional trainee working with a given Attending [19]. In terms of provider productivity, a previous study also revealed no significant difference in number of patients per hour seen by junior EM, senior EM, or off-service Residents [7]. The reasons that our study findings differ from previous reports include: 1) we balanced our samples of Attendings in both groups. Therefore, the changes in their operational performance can be compared at an individual Attending level. In contrast, other studies mainly assess the operational performance at a group level; 2) we used PDT instead of LOS for provider proficiency measurement which may help reduce the effects of other confounding factors (e.g., ED crowding). The use of propensity score matching could further minimize potential biases; 3) the study ED had 1:2 Attending-to-Resident ratio instead of 1:1 ratio staffing. This might affect attending operational performance more significantly than 1:1 teaching model, which could indirectly result in increased Attending productivity (i.e., more providers are available to see patients during the same time interval) with decreased efficiency (i.e., more trainees requiring more supervision per Attending per unit time), which differs from the previous report [7]. “

6. PLOS authors have the option to publish the peer review history of their article (what does this mean?). If published, this will include your full peer review and any attached files.

Do you want your identity to be public for this peer review? For information about this choice, including consent withdrawal, please see our Privacy Policy.

Reviewer #1: Yes: Marina Boushra

Reviewer #2: No

---

## [Decision Letter · Decision Letter 1]

10 Jan 2020

PONE-D-19-26595R1

Productivity, Efficiency, and Overall Performance Comparisons Between Attendings Working Solo Versus Attendings Working with Residents Staffing Models in an Emergency Department: A Large-Scale Retrospective Observational Study

PLOS ONE

Dear Dr. Wang,

Thank you for submitting your manuscript to PLOS ONE. After careful consideration, we feel that it has merit but does not fully meet PLOS ONE’s publication criteria as it currently stands. Therefore, we invite you to submit a revised version of the manuscript that addresses the points raised during the review process.

We would appreciate receiving your revised manuscript by Feb 24 2020 11:59PM. To enhance the reproducibility of your results, we recommend that if applicable you deposit your laboratory protocols in protocols.io, where a protocol can be assigned its own identifier (DOI) such that it can be cited independently in the future. For instructions see: http://journals.plos.org/plosone/s/submission-guidelines#loc-laboratory-protocols

We look forward to receiving your revised manuscript.

Kind regards,

Andrew Carl Miller

Academic Editor

PLOS ONE

Reviewers' comments:

Reviewer's Responses to Questions

**Comments to the Author**

1. If the authors have adequately addressed your comments raised in a previous round of review and you feel that this manuscript is now acceptable for publication, you may indicate that here to bypass the “Comments to the Author” section, enter your conflict of interest statement in the “Confidential to Editor” section, and submit your "Accept" recommendation.

Reviewer #1: All comments have been addressed

Reviewer #2: All comments have been addressed

2. Is the manuscript technically sound, and do the data support the conclusions?

Reviewer #1: Yes

Reviewer #2: Yes

3. Has the statistical analysis been performed appropriately and rigorously? 

Reviewer #1: I Don't Know

Reviewer #2: Yes

4. Have the authors made all data underlying the findings in their manuscript fully available?

Reviewer #1: Yes

Reviewer #2: Yes

5. Is the manuscript presented in an intelligible fashion and written in standard English?

Reviewer #1: Yes

Reviewer #2: Yes

6. Review Comments to the Author

Reviewer #1: (No Response)

Reviewer #2: It would be great if you were able to run additional analysis to address these 2 other questions:

Impact of EM vs. non-EM residents on attending productivity

Impact on attending and resident productivity when students are present

Line 87: students [5-7]., not students. [5-7].

Line 92: This is a loaded statement. The APPs were likely practicing independently, but the residents have to do more steps before disposition: present to attending, attending sees patient, then disposition. It's not really a fair comparison.

Line 164: Define how ED crowding was determined (i.e. NEDOCS score). Also state how overcrowding thresholds were selected, including references.

Line 175: please provide citation for STROBE

Line 204: “more elderly”: The statement is a little misleading. Although statistically significant, its not really clinically significant. The means were 48 vs 47 years of age.

Line 205: Fewer high acuity patients seen by solo attendings: This is a little misleading. Most places consider high acuity to be ESI 1&2 (not 1-3). The solo patients saw MORE ESI 1 (highest acuity) patients.

All high acuity patients are seen by an attending regardless of resident presence. There is a work-flow nuance here. On shifts with residents, there are likely fewer attendings, and on attending only shifts there is likely more attending coverage. The attending overseeing residents is likely to get more High Acuity patients because multiple residents are picking them up and presenting them. He/She can manage more at 1 time. On the attending only shifts, they are likely to divide up these patients, so any one attending will have fewer than if they were alone with residents. Please discuss this.

Table 1: what characterized ED overcrowding? Was it a NEDOCS score ≥ level 4?

For tables, please use superscript letters a for legends rather than symbols.

7. PLOS authors have the option to publish the peer review history of their article (what does this mean?). If published, this will include your full peer review and any attached files.

Reviewer #1: Yes: Marina Boushra

Reviewer #2: No

---

## [Author Response · Author response to Decision Letter 1]

20 Jan 2020

AUTHORS RESPONSE TO EDITORS AND REVIEWERS

As requested, we have included the original letter and comments with our point by point response in red colored font.

PONE-D-19-26595R1

Productivity, Efficiency, and Overall Performance Comparisons Between Attendings Working Solo Versus Attendings Working with Residents Staffing Models in an Emergency Department: A Large-Scale Retrospective Observational Study

PLOS ONE

Dear Dr. Wang,

Thank you for submitting your manuscript to PLOS ONE. After careful consideration, we feel that it has merit but does not fully meet PLOS ONE’s publication criteria as it currently stands. Therefore, we invite you to submit a revised version of the manuscript that addresses the points raised during the review process.

We would appreciate receiving your revised manuscript by Feb 24 2020 11:59PM. To enhance the reproducibility of your results, we recommend that if applicable you deposit your laboratory protocols in protocols.io, where a protocol can be assigned its own identifier (DOI) such that it can be cited independently in the future. For instructions see: http://journals.plos.org/plosone/s/submission-guidelines#loc-laboratory-protocols

• A rebuttal letter that responds to each point raised by the academic editor and reviewer(s). This letter should be uploaded as separate file and labeled 'Response to Reviewers'. 

• A marked-up copy of your manuscript that highlights changes made to the original version. This file should be uploaded as separate file and labeled 'Revised Manuscript with Track Changes'. 

• An unmarked version of your revised paper without tracked changes. This file should be uploaded as separate file and labeled 'Manuscript'.

We look forward to receiving your revised manuscript.

Kind regards,

Andrew Carl Miller

Academic Editor

PLOS ONE

Reviewers' comments:

Reviewer's Responses to Questions

Comments to the Author

1. If the authors have adequately addressed your comments raised in a previous round of review and you feel that this manuscript is now acceptable for publication, you may indicate that here to bypass the “Comments to the Author” section, enter your conflict of interest statement in the “Confidential to Editor” section, and submit your "Accept" recommendation.

Reviewer #1: All comments have been addressed

Reviewer #2: All comments have been addressed

2. Is the manuscript technically sound, and do the data support the conclusions?

Reviewer #1: Yes

Reviewer #2: Yes

3. Has the statistical analysis been performed appropriately and rigorously? 

Reviewer #1: I Don't Know

Reviewer #2: Yes

4. Have the authors made all data underlying the findings in their manuscript fully available?

Reviewer #1: Yes

Reviewer #2: Yes

5. Is the manuscript presented in an intelligible fashion and written in standard English?

Reviewer #1: Yes

Reviewer #2: Yes

6. Review Comments to the Author

Reviewer #1: (No Response)

Reviewer #2: It would be great if you were able to run additional analysis to address these 2 other questions:

Impact of EM vs. non-EM residents on attending productivity

Response: Thanks for reviewer’s valued comments. Our attending physicians are staffed to work with one senior EM resident and one junior EM resident or non-EM resident during the same working hours. In this study, we reported attending productivity using number of new patients per hour and this is a mixture of new patients seen by either one senior EM resident + one junior EM resident, or one senior EM resident + one non-EM resident. Therefore, we are unable to do a separate report to see the impact of EM vs. non-EM residents on attending productivity. However, we did another separate analysis to only determine the resident productivity and we found that non-EM resident saw an average of 1.7 patients/hour, similar to junior EM resident (1.8 patients/hour). Based on this evidence, we assume that attending productivity have no significant differences when attending working with 1) one senior EM resident and one junior EM resident versus 2) one senior EM resident and one non-EM resident. We revised our manuscript and added a discussion to our limitation. (Page 16, Line 344-6)

Impact on attending and resident productivity when students are present

Response: Again, thanks for reviewer’s valued comment. In the study ED, medical students see patients together with our senior EM residents, but they are not allowed to do any documentation in the EMR system during study period. Senior EM residents report patients to the attending and do the medical documentations. This could potentially affect senior EM resident productivity to a certain level. Unfortunately, due to the lack of medical students’ data, we were unable to report this impact on attending productivity when students are present. We revised our manuscript and added to our limitation to address this. (Page 16, Line 344-6)

Line 87: students [5-7]., not students. [5-7].

Response: Yes, it is revised. (see Page 4, Line 87)

Line 92: This is a loaded statement. The APPs were likely practicing independently, but the residents have to do more steps before disposition: present to attending, attending sees patient, then disposition. It's not really a fair comparison.

Response: Thanks for reviewers’ valued comments and sorry for the confusion. This is truly what we want to address: residents in training might be less efficient than other providers not in training statuses (such as APPs). We revised this sentence and make it clearer. (see Page 4, Line 90-93). 

Line 164: Define how ED crowding was determined (i.e. NEDOCS score). Also state how overcrowding thresholds were selected, including references.

Response: Yes, we defined ED crowding using NEDOCS in Appendix (Appendix-1.2) and also added the thresholds of not-crowding, crowding, and overcrowding separately with references (see Appendix with separate references). Details are as the followings:

Appendix 1.2: Detail Variables Explanation

ED Length of Stay (LOS) is defined as the time in minutes beginning at the point the patient is initially arrived and registered into the EMR at the ED and ending at the point that the patient physically leaves the ED indicating closure of that specific encounter. 

Provider-to-Disposition time (PDT) is defined as the time interval documented in the EMR beginning at the point when the patient is initially seen and evaluated by a provider and ending at the point when the disposition decision is made. 

The number of new patients per hour is defined as the number of new patients assigned to individual providers within a one-hour block (e.g., 0200 to 0259). 

ED crowding is measured using the NEDOCS score (National Emergency Department Over-Crowding Study) upon each patient’s arrival to the ED (see below). The definitions of these variables are consistent with those previously published (Welch SJ, Asplin BR, Stone-Griffith S, Davidson SJ, Augustine J, Schuur J: Emergency department operational metrics, measures and definitions: results of the Second Performance Measures and Benchmarking Summit. Ann Emerg Med 2011, 58: 33-40.). NEDOCS score equals or less than 100 (NEDOCS≤100) is considered ED not crowding, NEDOCS score less than 140 but greater than 100 (100<NEDOCS<140) is considered ED crowding, and NEDOCS score equal or greater than 140 (NEDOCS≥140) is considered ED overcrowding.

Detail Explanation of NEDOCS Score Calculation

Variables Definition

NEDOCS = 85.8T + 600B + 5.64W + 0.93L + 13.4C -20

T The total number of ED patients collected divided by the number of licensed beds at the time a score is calculated

B The number of admitted patients/number of hospital beds at the time a score is calculated

W Longest wait time in hours for patients in the waiting room at the time a score is calculated

L Longest time in hours since registration among boarding patients at the time a score is calculated

C Number of critical care patients at the time a score is calculated. Typically, this is a site-specific variable which usually refers to patients that require one-to-one nursing care. In the study ED, critical care patients are defined as ICU patients and ICU consulted patients including but not limited to patients on mechanical ventilators, receiving tPA, diagnosed with septic shock, critical trauma patients, and patients requiring conscious sedation at the time a score is calculated, etc.

Line 175: please provide citation for STROBE

Response: Yes, it is added. (see Page 8, Line 176)

Line 204: “more elderly”: The statement is a little misleading. Although statistically significant, its not really clinically significant. The means were 48 vs 47 years of age.

Response: Yes, we deleted “more elderly” since there was no clinically significant value for such comparisons. We revised as the following: “It was noted that the attending working solo group saw slightly more high-acuity (ESI-1) patients”. (Page 9, Line 205-206)

Line 205: Fewer high acuity patients seen by solo attendings: This is a little misleading. Most places consider high acuity to be ESI 1&2 (not 1-3). The solo patients saw MORE ESI 1 (highest acuity) patients.

All high acuity patients are seen by an attending regardless of resident presence. There is a work-flow nuance here. On shifts with residents, there are likely fewer attendings, and on attending only shifts there is likely more attending coverage. The attending overseeing residents is likely to get more High Acuity patients because multiple residents are picking them up and presenting them. He/She can manage more at 1 time. On the attending only shifts, they are likely to divide up these patients, so any one attending will have fewer than if they were alone with residents. Please discuss this.

Response: Sorry for the confusion. We revised the result and addressed more accurately as the following: “It was noted that the attending working solo group saw slightly more high-acuity (ESI-1) patients”. This could possibly be due to: 1) the daily variety of high acuity patients present at the study ED (e.g. slightly more ESI-1 patients present during non-resident shifts); and 2) the exclusion of the APPs in this study. During the ED non-resident time (16h/week), an extra APP was scheduled to work. However, this study excluded APPs who usually only saw low-acuity patients. This is the reason that propensity score matching comparison was used in the study to minimize these confounders. We also revised and addressed it in our discussion section. (See Page 13, Line 279-283).

Table 1: what characterized ED overcrowding? Was it a NEDOCS score ≥ level 4?

For tables, please use superscript letters a for legends rather than symbols.

Response: Yes, ED overcrowding was addressed in detail in Appendix. NEDOCS score >140 is considered overcrowding. In addition, we revised and used superscript letters for legends (See revised Table 1 and 2). 

7. PLOS authors have the option to publish the peer review history of their article (what does this mean?). If published, this will include your full peer review and any attached files.

Do you want your identity to be public for this peer review? For information about this choice, including consent withdrawal, please see our Privacy Policy.

Reviewer #1: Yes: Marina Boushra

Reviewer #2: No

---

## [Editor Report · Decision Letter 2]

23 Jan 2020

Productivity, Efficiency, and Overall Performance Comparisons Between Attendings Working Solo Versus Attendings Working with Residents Staffing Models in an Emergency Department: A Large-Scale Retrospective Observational Study

PONE-D-19-26595R2

Dear Dr. Wang,

We are pleased to inform you that your manuscript has been judged scientifically suitable for publication and will be formally accepted for publication once it complies with all outstanding technical requirements.

With kind regards,

Andrew Carl Miller

Academic Editor

PLOS ONE

Additional Editor Comments (optional):

The authors appropriately addressed each of the reviewer comments.

---

## [Editor Report · Acceptance letter]

28 Jan 2020

PONE-D-19-26595R2 

Productivity, Efficiency, and Overall Performance Comparisons Between Attendings Working Solo Versus Attendings Working with Residents Staffing Models in an Emergency Department: A Large-Scale Retrospective Observational Study 

Dear Dr. Wang:

I am pleased to inform you that your manuscript has been deemed suitable for publication in PLOS ONE. Congratulations! Your manuscript is now with our production department. 

With kind regards,

on behalf of

Dr. Andrew Carl Miller 

Academic Editor

PLOS ONE